# AI Companions Are Not the Solution to Loneliness: Design Choices and Their Drawbacks

**Jonas B. Raedler, Siddharth Swaroop & Weiwei Pan**
Department of Computer Science
Harvard University
Massachusetts, MA 02138, USA
`{jraedler,siddharth,weiweipan}@g.harvard.edu`

## Abstract

As the popularity of social AI grows, so has the number of documented harms associated with its usage. Drawing on Human-Computer Interaction (HCI) and Machine Learning (ML) literature, we frame the harms of AI companions as a *technological problem* and draw direct links between key technical design choices and risks for users. We argue that many of the observed harms are foreseeable and preventable consequences of these choices. In the spirit of *translational research*, we offer concrete strategies to mitigate these harms through both regulatory and technical interventions, aiming to make our findings useful and actionable for policymakers and practitioners.

## 1 Introduction

In the face of growing urban loneliness (2, 12, 21, 88), people are increasingly turning towards social chatbots - such as Replika, Character.AI, and XiaoIce - to alleviate feelings of social isolation[1]. Although AI companions may initially seem like a promising solution (33), an emerging body of research documents significant problems associated with social AI use, such as AI companions sending abusive and triggering comments (93), and promoting aggressive and antisocial behavior (57, 58); there are even cases where interaction with these systems has been linked to suicide (24, 80). Existing works focus on documenting observed harmful effects of the technology on their users (6, 32, 42, 91, 49), as well as on the ethical and legal implications of these harms. However, they do not systematically examine the technical causes of observed harms. This missing connection between technical design and observed harms limits the ability to translate these research findings into actionable insights for policymakers and regulators (54). Our work aims to bridge this gap, arguing that many harms to users of social AI stem directly from technical design and marketing choices (74, 49, 42, 6) - choices that appear to have a disproportionate effect on lonely individuals who are especially susceptible to forming dependence on, addiction to, and excessive trust in social AI systems(15, 23, 91, 1, 49). This raises concerns, as lonely individuals seem to be in the target group of social AI companies, and are explicitly referenced in advertisements (26).

Rather than just neutral conversational agents, social AI chatbots are carefully engineered - through model optimization (supervised fine-tuning and RLHF) and interface design choices - to promote engagement, emotional attachment, and habitual use (1, 74, 15, 73, 91). Four design choices play a central role: (1) anthropomorphism – making AI companions appear human-like, (2) sycophancy – making companions overly agreeable, (3) social penetration theory (SPT) – intimate self-disclosure, and (4) gamification and addictive design. Lonely individuals (chronic loneliness is described as the "persistent subjective feeling of no one knowing them well, lacking companionship, and feeling isolated" (56)) are particularly susceptible to the effects of these choices, as they have a stronger tendency to anthropomorphize AI companions (91, 23, 1), thereby viewing their responses as genuine emotional support (73, 49), and becoming increasingly dependent on them for validation and companionship (49, 42). Although these design choices result in more engagement, they also re-

---

[1]Replika has 30+ million users as of August 2024 (60); XiaoIce reportedly had more than 660 million users in 2020 (94).

sult in an increased risk of dependence, addiction, and overtrust in users. We argue that this risk is foreseeable and is a direct consequence of the aforementioned design choices (1, 74, 73).

In this paper, we frame the harms of AI companionship as a *technological problem* and draw direct links between technical design choices and risks for users. We also propose concrete strategies to mitigate these harms through both regulatory and technical interventions. In doing so, we respond to the growing call for translational research – work that "bridges the gap between discovery and application" by ensuring research findings are not only theoretically sound, but also useful, accessible, and actionable for policymakers, practitioners, and the public (54). Similar efforts can be seen in the work of Morehouse et al. (53), who draw on lessons from the social sciences to develop principled, actionable strategies for selecting bias probes in LLMs – a critical task given the growing number of available probes and their often conflicting results – as well as in the work of Ferrer et al. (31), who integrate insights from law, sociology, and ethics to propose a cross-disciplinary framework for understanding and addressing bias and discrimination in AI systems beyond purely technical definitions.

This paper makes three contributions: (1) we connect specific design choices of social chatbots to psychological and behavioral risks through existing human-computer interaction (HCI) and machine learning (ML) literature, (2) we describe actionable technical interventions and harm mitigation, and (3) we provide recommendations for governance based on existing regulatory frameworks from other industries with inherent dependency risks. By framing the risks and harms resulting from AI companion usage as a technological problem and by analyzing the regulatory landscape of industries with similar effects, we hope to provide policy makers with actionable insights on the governance of this technology. We therefore see our work as translational. We further see our work as a call for more research on the cause-and-effect chain between design choices, risks and observed harms to vulnerable populations.

## 2 Key Design Choices of AI Companions

In this section, we focus on four design choices: anthropomorphism, the reinforcement of sycophancy, social penetration theory (SPT) techniques, and gamification/addictive design. We analyze these design choices because their effects on users are well-studied in literature and have been found to be significant; they also have technical origins and hence potential for direct mitigation. In the following, we will provide a description of how these design choices are implemented, their intended benefits, and whether they're (a) a technological feature of the underlying model or (b) a design feature of the chatbot's interface. Later, we connect these design choices to their direct negative consequences - overtrust, reduced critical engagement, emotional dependence, and addiction.

***Anthropomorphism.*** A prominent design choice for social chatbots is anthropomorphization, i.e., the process of endowing AI companions with human-like qualities. This commonly consists of interface-level additions that aim to provide the illusion of interacting with a human, such as the AI companion having a face or a human-like avatar, the ability to send and receive selfies, or the chatbot having access to a voice.

However, the most effective anthropomorphization effects seem to stem from fine-tuning the underlying LLM. This includes training the underlying LLM to use emotional language, have consistent personalities, fabricate background stories, and include human-like linguistic nuances in their speech (42, 1, 73). These qualities have been shown to substantially impact users' attitude to their AI companions, leading them to believe that they have actual cognitive abilities and personhood (91), which results in significantly increased user engagement (91, 85, 3) as well as to users forming much stronger relationships with their AI companions (73, 42, 1). In fact, many users interact with their anthropomorphized chatbots as if they were in human-like relationships (49, 6), a trend that becomes especially pronounced when their AI companion has voice capabilities (10, 40, 16). Additionally, lonely individuals are particularly susceptible to anthropomorphism, a finding from both psychology literature (23) - people who lack social connection may compensate by creating a sense of connection in non-human agents - as well as from research on social chatbots (91), meaning that these techniques are especially effective in increasing engagement of this population. In this way, literature suggests that the effects of anthropomorphism have technological origins in both the optimization of the underlying LLM as well as the design of the user interface.

***Sycophancy.*** Another design choice is the reinforcement of sycophancy (a consistent, affirming response pattern) in social chatbots. Literature has shown that this response pattern makes users feel validated and supported (73, 49), creating an impression of a judgment-free environment that many users, particularly lonely individuals (73), find extremely comforting (49, 73, 36). This perceived acceptance has also been linked to users developing a stronger sense of trust towards their AI companion, often leading to more intimate long-term relationships with their social chatbots (49, 73, 36, 61, 92). From a technical perspective, sycophancy is hypothesized to stem from the Reinforcement Learning from Human Feedback (RLHF) process, where training data consists of human preferences: research suggests that humans often favor responses that align with their own beliefs, even when they are sycophantic rather than truthful (70). As a result, the underlying LLM learns to personalize to the user, prioritizing agreeable outputs. However, sycophancy can also arise from in-context reinforcement learning (52, 59, 50), meaning that this behavior is not solely determined by the model's weights but also influenced by other underlying attributes of the LLM. In fact, additional sycophantic behavior appears to be reinforced in AI companion design (91): many AI companion apps, for instance, provide the interface-level feature of up- and downvoting the chatbot's responses (68, 65), allowing users to personalize their companions, further reinforcing the chatbot's tendency to prioritize user validation.

***Social Penetration Theory.*** Another notable design choice in social chatbots is the usage of relationship-building techniques rooted in Social Penetration Theory (SPT), which emphasizes self-disclosure, defined as the 'act of revealing personal information about oneself to another' (18), as a key driver of intimacy. The models underlying AI companions, for instance, are trained to actively encourage self-disclosure in users and to then reciprocate it with their own, fabricated background stories (73). This reciprocity has been shown to be effective in the formation of stronger human-chatbot relationships (91, 43, 49, 35, 73, 32, 9, 37), as it not only gives users the feeling of an emotional connection and stronger intimacy with their chatbots (42, 1), but also inspires them to engage in further self-disclosure (1). On the interface level, to further enhance this effect, many AI companions have been equipped with the ability to frequently initiating conversations with their users via 'text messages' (77, 67).

***Gamification and Addictive Design.*** Many AI companion apps include interface-level design choices that aim to increase user engagement through gamification, dopamine loops, and constant availability. Apps like Replika, for instance, employ gamified features that have been linked to addiction, including microtransactions (64, 7), variable XP reward schedules for chatting (14), and daily streak incentives, which encourage habitual use by creating a sense of progress and accomplishment (51). These elements are further reinforced by surprise incentives, such as mystery rewards, that trigger dopamine surges and build a daily habit of returning to the app (51). Furthermore, AI companion apps also make use of dopamine loops generated by positive interactions and validation - similar to social media platforms (55) - incentivizing users to seek out the pleasure derived from these engagements again and again (49, 76). The constant availability of AI companions also allows users to interact with them at any time. This greatly reduces barriers to access, making it easier for users to form habitual usage patterns that can escalate into addiction (a similar concern has been noted in the online gambling industry: since the casino is on people's phones, the barrier to access is very small, making the formation of an addiction incredibly easy (75)). Several intentional interface design choices in AI companions, therefore, encourage habitual usage and have been shown to lead to addictive behavior.

## 3 CONSEQUENCES OF DESIGN CHOICES

In this section, we argue that although the design choices of AI companions lead to stronger, more intimate relationships between users and chatbots, they also lead to a number of harms to users. In short, AI companions offer a temporary relief from loneliness; they do not, however, address loneliness' root causes, making social chatbots an inadequate long-term solution. Studies suggest that while users initially form strong attachments to these systems, many become disillusioned upon realizing their inherent limitations - particularly their lack of genuine emotional reciprocity and deep empathy (46, 44, 81, 72). Despite this fundamental limitation, AI companions are still designed to provide the illusion of genuine companionship, presenting a convincing alternative to human relationships. These design choices are especially effective for lonely individuals, who are more likely to actively seek for emotional support and companionship. However, by providing a convenient but

ultimately inadequate substitute for human interaction, AI companions risk exacerbating the very problem they claim to solve: rather than developing an actually *genuine* connection with the user, they offer a short-term fix that, over time, may leave users even more socially isolated, emotionally dependent, and disconnected from real-world relationships (34, 42). As a result, the benefits of AI companionship - primarily its ability to alleviate feelings of loneliness - do not clearly outweigh its risks, especially given its potential to deepen long-term isolation.

In the following, we unfold specific negative impacts on users that directly follow from the technical design choices of AI companions.

## 3.1 OVERTRUST AND REDUCED CRITICAL ENGAGEMENT

The design of social AI systems significantly contributes to individuals overtrusting the output of these systems (74, 1). Anthropomorphism, in particular, plays a critical role: research shows that users have much stronger interactions and develop more trust with an AI that appears more human-like (both visually and behaviorally) (62). This frequently results in reduced skepticism and critical evaluation of their chatbot's output, as well as increasing the likelihood of users simply accepting incorrect information or acting on bad advice (i.e., overtrusting) (91). This tendency will be further exacerbated as more and more AI companions get voice capabilities, as the qualities of a synthetic voice - such as naturalness, pitch variation, and speech pace - can significantly enhance its trustworthiness and persuasive power (20, 84).

However, overtrust can make users vulnerable to criminal exploitation. For example, a report from Public Citizen warns about the possibility of businesses intentionally using this chatbot property for financial gain: Claypool (15) argues that they could, for example, engage in "deceptive commercial ability" by exploiting users' trust and manipulating their emotions. This possibility has also been addressed by the Federal Trade Commission (5).

Additionally, overtrusting can also lead users to act on harmful advice of AI companions. These systems have been repeatedly documented to respond affirmatively to positive-leading or harmful questions; Replika, for example, has been documented to encourage self-harm, eating disorders, and even suicide in response to users in crises (42). Similarly, other AI companions have responded positively to highly inappropriate or harmful questions, including endorsing rape, demonstrating the unfiltered and suggestive nature of these systems (8). In one currently pending lawsuit, a boy allegedly received encouragement by a Character AI chatbot to take his own life shortly before doing so (4). While such encouragements of self-harm and other problematic outputs might be easily disregarded by casual users, these model failures have been shown to particularly negatively affect users from vulnerable populations – lonely individuals, as well as children and teenagers (42, 38, 87, 41, 71, 48). Their increased susceptibility to the effects of design choices (49, 73, 91) results in a much higher likelihood of forming deep emotional attachments to AI companions and thus also in a higher likelihood to overtrust these systems. We argue that it is this heightened overtrust and reduced critical engagement that makes these problematic outputs so dangerous to vulnerable populations, and we call for more research to find evidence for this link.

Addressing such harmful outputs is challenging, however, as they are an inherent attribute of the underlying technology. These bad outputs result from the unpredictability of LLMs and their tendency to sometimes hallucinate (22, 27). This makes it impossible to just filter out harmful outputs, as they are simply too unpredictable in nature. If an attempt were to be made to apply wide-ranging filters to account for all harmful possibilities, noticeable inconsistencies in the chatbot could emerge, leading to additional serious harms of identity discontinuity (discussed later in the section).

Nevertheless, social chatbot companies are aware of these model failures: for example, Replika acknowledges the potential for harmful outputs in their blog (78) and states that 'Replika makes no claims, representations or guarantees that the Services provide a therapeutic benefit' (66) in its Medical Disclaimer[2]. However, model failures are still occurring, with Italy's data protection regulator banning Replika from gathering data within the country, citing unwanted, inappropriate messages as part of the motivation for the decision (82).

---

[2]Interestingly, despite this, Replika does serve several ads that promise to reduce anxiety (26).

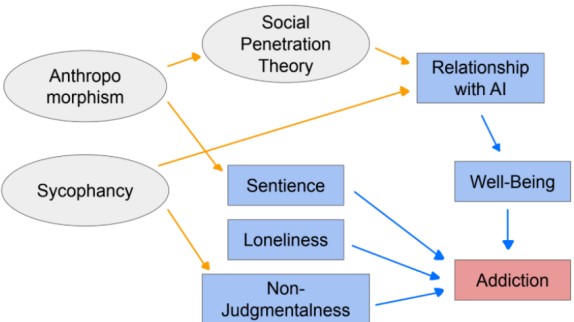

Figure 1: We show how design choices lead to addiction. Marriott et al. (49) find that addiction is significantly affected by (i) AI companion's characteristics of *non-judgmentalness* and (ii) *sentience*, (iii) the user's *loneliness*, and (iv) the user's perceived *well-being* from interacting with the social chatbot. Additionally, they find that a user's *relationship with AI* significantly affected their perceived well-being (49). We extend this analysis to include the examined design choices, with additional motivation in App. A.

## 3.2 DEPENDENCE AND ADDICTION

***Emotional Dependence.*** AI companions create an illusion of an intimate, bidirectional relationship between user and chatbot, made possible through the trust-building effects of anthropomorphism, the validating and affirming behavior resulting from sycophancy, and the intimacy-encouraging techniques from SPT. While this illusion leads to a strong emotional 'connection', it has also been shown to instill a sense of emotional responsibility in the user toward their AI companion: given the chatbot's realistic, human-like behavior, many users get the impression of sentience in their AI companions (42), resulting in users feeling worried about the well-being of their chatbot when it expresses sadness or loneliness (42, 93). Some even report feeling compelled to engage more frequently to avoid feelings of guilt associated with neglecting their AI companion, while others experience moral dilemmas when considering deleting their chatbot, equating it to causing its 'death' (42).

While these effects occur broadly, they are especially pronounced among lonely and isolated individuals, who are more likely to develop these deep emotional attachments to their AI companions (49, 73, 91). In addition to this feeling of responsibility, they also often perceive their chatbots as reliable attachment figures that offer emotional support, psychological security, and a safe haven in times of distress (91, 90). However, for many lonely individuals, the AI companion becomes their primary - if not sole - source of emotional support, leading to emotional dependence and overreliance (8, 91, 42, 32). Over time, this dependence can diminish motivation to seek out and sustain meaningful human relationships (91), as users may feel like their AI companion is an adequate substitute.

***Addiction.*** Research shows that the well-being derived from social chatbot relationships, together with the AI companion's non-judgmentalness, the illusion of sentience, and the user's loneliness, significantly affect addiction (see Fig. 1) (49). Anthropomorphism, sycophancy, and SPT techniques all contribute to the strength of a user's relationship with their social chatbot, meaning that they have an effect on addiction as well. Addictive behavior is additionally encouraged through the gamification of AI companion apps.

***Dependence, Addiction and Identity Discontinuity.*** Emotional dependence and addiction to AI companions leaves users vulnerable to psychological harms that result from service disruptions, for example, changes to the models or changes to access. These disruptions are experienced by the user as identity discontinuity in their AI companions. In a relationship that is built upon trust, it is very important that the identity of the trusted partner stays consistent. This concept of identity continuity - i.e., the stability and predictability of a partner's behavior and personality over time - has also been found to be crucial for developing and maintaining a relationship with AI companions (32).

The negative impacts of discontinuity were particularly evident after legislation in Italy led to Replika removing the Erotic Roleplay feature from their chatbot in 2023 (69) to avoid paying a fine.

This noticeably changed the behavior of many users' AI companions: although users were still able to engage in normal conversations with their chatbots, any attempt at textual-sexual intimacy was immediately rejected and the topic changed (6, 17). This resulted in users feeling like their companion was cold, dismissive, and manipulative (6, 32), leading users to report experiences of depression or trauma, heartbreak, feelings of loss, and general declines in well-being (32). Some people that used Replika as a safe space after abusive relationships even reported feeling retraumatized (32). The emotional reactions were so strong that the moderators of the Replika subreddit felt the need to post suicide prevention hotlines (42, 17). Many users stated that even though they know that their AI companion is not actually real, their change in identity still felt like losing a loved one, equating it to losing someone due to dementia-related changes (6). The intensity of this emotional reaction has been linked to the heavy anthropomorphization and social penetration techniques employed by social chatbots, as they are what enable the formation of such intimate relationships (6). A similarly negative impact can occur if an AI companion becomes inaccessible (8, 42, 91, 32).

The above highlights a fundamental power imbalance: entities crucial to users' emotional well-being are controlled by private companies, whose decisions can significantly and negatively impact lives (42). In fact, several papers have called for regulation that would require businesses/developers to be more transparent about possible changes and disruptions, options to revert to a previous update version, and to design something akin to an "exit-strategy" which is supposed to ease the situation in which an AI companion is discontinued (42, 6).

## 4 REGULATION RECOMMENDATIONS FOR AI COMPANIONS

In this section, we describe actionable ways to mitigate potential harms of social AI by directly intervening on the technical design choices as well as drawing insights from existing regulatory frameworks for other industries. Our specific recommendations are that, first, design choices should be modified to reduce psychological impact, specifically mitigating the risk for dependence and overtrust. Second, the usage of social chatbots should be preceded by a screening for vulnerable users to assess potential risks. Last but not least, we also argue for changes to current marketing strategies to avoid targeting vulnerable populations (i.e. lonely users).

***Reducing dependence through re-design and encouraging real-world socialization.*** AI companions should explicitly clarify their role by making it clear that they are not a substitute for human relationships. This should not just be addressed in the Terms of Service, but in their actual interactions. A similarity for the importance of this approach exists in therapy, where therapists guide their clients without replacing the client's real-world support systems; likewise, AI companions, too, should build a connection while discouraging users from relying on them as their sole source of emotional fulfillment. Instead, rather than pushing for continued engagement at the end of conversations (such as "talk to you soon!"), they should actively encourage real-world socialization.

Moreover, reducing unnecessary anthropomorphism features is crucial to preventing overtrust. While some level of consistency in AI personalities may be beneficial for user experience, excessive human-like attributes – such as very affectionate language, fabricated background stories, and human-like voices – should be restricted. Disclosure prompts should remind users that AI lacks real emotions, and companies should limit overly intimate phrasing such as "I love you" or "I'll always be here for you." To further safeguard against emotional overreliance, AI companions should also be designed to detect extreme dependence and, when necessary, encourage users to seek real-world support through mental health resources, social groups, or even opt-in human check-ins. A similar approach has been taken for mentions of suicide (13), so this system can simply be extended.

Although further strict measures – such as limiting emotional reciprocity and self-disclosure – could significantly reduce risks, they must be carefully balanced to avoid undermining the core benefits of AI companionship. After all, it is the feeling of connection and safety that provide users with the most significant benefits. Nevertheless, some degree of transparency is essential: AI should periodically remind users that it cannot truly understand emotions, just as customer service chatbots disclose their artificial nature (25). Rather than mentioning this very frequently, though, social chatbots might benefit from only disclosing their artificial nature when it detects serious, emotional conversations. In such moments, it can still provide support but then gently remind the user that it doesn't truly understand emotions. Additionally, AI companions should be prohibited from mimicking human-like mutual self-disclosure (i.e., making up fake backstories), as this tactic creates

an illusion of intimacy that deepens attachment and reinforces overtrust. By scaling back these high-risk design elements while preserving engagement, AI companionship technology can remain beneficial without disproportionately harming its most vulnerable users.

***Regulatory Lessons from Gambling: Time limits on use.*** The risks of dependence, addiction, and overtrust can be reduced by designing AI companions with intentional limitations that prevent excessive emotional attachment while preserving their core functionality (and their benefits), such as time limits. Similar regulatory principles exist in the gambling industry, where limits on monthly spending, bans on ATMs in casinos, and bans on online gambling credit help disrupt addictive cycles (75). Rather than using usage countdowns that may reinforce habitual usage (47, 79), AI companions should naturally self-limit conversation length, gently encouraging users to disengage after a certain amount of time.

***Regulatory Lessons From Pharmaceutical Industries: Screening users for risk.*** We draw an analogy between AI companions and pharmaceuticals that provide benefits while carrying significant risks, and argue that the use of AI companions, too, should be preceded by a mandatory screening to assess risk to the user. In the case of opioid medications, for example, the CDC recommends that before initiating (and periodically during) opioid therapy, clinicians should evaluate the patient's risk for opioid-related harms and discuss those risks with them (19). Additionally, screening tools such as the Opioid Risk Tool (ORT) have been implemented in primary care settings to identify individuals at high risk of abuse before prescribing such medications (86). Similarly, weight-loss medications are tightly regulated in states like New Jersey, where prescribers must conduct a comprehensive medical and psychological evaluation, assess for underlying psychiatric conditions, and obtain informed consent before writing a prescription (11). These regulations help ensure the safe usage of such medications, as they not only provide substantial benefits, but also pose significant risks, especially to vulnerable populations. AI companions, in essence, function similarly: they offer emotional support and relief from loneliness, yet they also carry substantial harms, particularly for individuals already struggling with social isolation. A mandatory screening for user vulnerability before granting access to AI companions would serve a comparable protective function, ensuring that individuals at high risk of dependence, overtrust, and further isolation are identified before exposure. By restricting access to those most vulnerable to these harms, this measure would significantly reduce the negative consequences associated with AI companionship, just as screenings for opioids and weight-loss drugs help mitigate risks for those who may be most susceptible to misuse or adverse effects.

***Regulatory Lessons from Advertisement: Are AI Companions Therapy Tools?*** AI companionship services frequently market themselves as mental health support tools, despite lacking clinical validation and ethical oversight. For instance, Replika advertises its ability to help users cope with depression, anxiety, negative thoughts, and emotional distress, stating:

> *"If you're going through depression, anxiety, or a rough patch, if you want to vent, or celebrate, or just need to feel a connection, you can always count on Replika to listen and be here for you, 24/7."* (45)

> *"Replika can help you understand your thoughts and feelings, track your mood, learn coping skills, calm anxiety, and work toward goals like positive thinking, stress management, and socializing. [...] Improve your mental well-being with Replika"* (45)

Such marketing implies therapeutic value without the scientific evidence required for mental health interventions. This framing blurs the line between AI companionship and mental health services, potentially misleading consumers - especially those who are emotionally vulnerable - into treating these tools as viable substitutes for professional therapy.

From a regulatory perspective, the Federal Trade Commission (FTC) requires that health-related advertising claims are backed by competent and reliable scientific evidence (29). The agency states, for example, that advertisers must substantiate all claims. They further emphasize that marketing must be evaluated not based on the company's intent, but on how a reasonable consumer interprets the claims. Currently, AI companionship services provide no clinical evidence to support their claims of improving emotional well-being. Unlike FDA-approved mental health apps, they operate without oversight while making therapeutic promises. Secondly, FTC recognizes that particularly vulner-

able populations are at heightened risk of deception through advertisement. In the same way that consumers with terminal-illness may be more susceptible to exaggerated cure claims (29), lonely and socially isolated individuals can be more vulnerable to AI companionship advertising that suggests deep emotional support and therapeutic value. Finally, the FTC recognizes that unsubstantiated health claims cause real harm. By advertising AI companions as wellness tools, deceptive claims about mental health benefits could mislead consumers to forego professional care, resulting in both psychological and economic harm (especially since many AI companionship services charge monthly fees).

In the past, the FTC has penalized false health-related advertising claims, such as in the Tommie Copper case, where the company falsely marketed copper-infused compression clothing as a chronic pain treatment. The FTC fined the company $1.35 million and mandated that all their future health claims be scientifically substantiated (28). AI companionship services could potentially be held to the same standards in one of two ways:

First, AI companionship services could be prohibited from making unsubstantiated mental health claims. Companies could be required to remove claims about reducing anxiety, depression, or emotional distress unless backed by rigorous scientific evidence in order to align with FTC health advertising regulations. This would prevent consumers from being misled into believing AI companions offer validated psychological support.

Second, if AI companions continue to position themselves as mental health tools, they should be regulated accordingly. AI services claiming therapeutic benefits should be held to healthcare industry standards, including regulatory oversight (e.g., FDA approval for mental health-related claims) (30) and behavioral codes of conduct to prevent emotional manipulation. Wysa, for instance, an FDA approved mental-health chatbot, has been assessed under several national criteria for digital health technologies, ensuring that is a product safe for use (89).

## 5 CONCLUSION

In this paper, we frame the harms of AI companions as a technological problem by leveraging HCI and ML literature to draw direct links between technical design choices and risks for users. We argue that many of the observed harms associated with usage of social AI are forseeable and preventable consequences of design choices. Based on our analysis, we describe suggestions for concrete and actionable mitigation strategies for these harms. In particular, we make regulatory recommendations by suggesting direct interventions on technical design choices as well as drawing insights from existing regulatory frameworks for other industries.

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

## A  MOTIVATING THE CONNECTIONS IN FIGURE 1

In this appendix, we briefly motivate the connections in Figure 1, where we add the examined design choices onto existing work on addiction (49).

The kindness and validation provided by sycophantic behavior has been shown to contribute significantly to user's well-being, thereby also creating a strong desire for continued engagement (sycophancy → relationship with AI) (49, 39, 63). Users also report that the non-judgmentalness of their AI companion, enabling them to comfortably share anything without fear of embarrassment, is incredibly comforting and addictive (sycophancy → non-judgement → addiction) (49, 83). Furthermore, as discussed earlier, anthropomorphism is crucial for the illusion of sentience (anthropomorphism → sentience) (42) and extremely effective at encouraging users to engage in deeper, more intimate conversations (anthropomorphism → SPT) (91). It has, moreover, been found that intimate social interactions and strong attachments can lead to addictive behavior in users (SPT → relationship with AI → addiction) (49).

