# OpenReview forum: "AI Companions Are Not The Solution To Loneliness: Design Choices And Their Drawbacks"
_ICLR.cc/2025/Workshop/BuildingTrust — BuildingTrust_

### Official Review · Reviewer_xAPf · 2025-02-28
**A good paper**

**Rating:** 7
**Confidence:** 3

**Review:**

The paper puts into perspective the risks of AI companions for vulnerable populations and how most of them are by design and could be mitigated. I think this paper is very nice and insightful. I cannot give a higher score only because I am not familiar enough with the literature on this subject to say if it is an exceptional good paper.

---

### Official Review · Reviewer_cmQF · 2025-03-01

**Rating:** 9
**Confidence:** 4

**Review:**

This paper is an extremely important and well-written contribution that highlights how technological design choices we (AI researchers) make are directly related to downstream risks and harms for vulnerable users, specifically lonely people. I believe every LLM researcher building a system intended to be used by a human would benefit from reading this paper.

---

### Decision · Program_Chairs · 2025-03-04

**Decision:**

Accept

**Comment:**

This paper highlights how AI design choices impact vulnerable users, particularly lonely individuals, and emphasizes the risks of AI companions. It offers valuable insights for LLM researchers, suggesting that many risks are inherent by design but can be mitigated.